# Identification of IncA Plasmid, Harboring *bla*_VIM-1_ Gene, in *S. enterica* Goldcoast ST358 and *C. freundii* ST62 Isolated in a Hospitalized Patient

**DOI:** 10.3390/antibiotics12121659

**Published:** 2023-11-25

**Authors:** Alessandra Piccirilli, Sascia Di Marcantonio, Venera Costantino, Omar Simonetti, Marina Busetti, Roberto Luzzati, Luigi Principe, Marco Di Domenico, Antonio Rinaldi, Cesare Cammà, Mariagrazia Perilli

**Affiliations:** 1Department of Biotechnological and Applied Clinical Sciences, University of L’Aquila, 67100 L’Aquila, Italy; sascia.dimarcantonio@graduate.univaq.it (S.D.M.); mariagrazia.perilli@univaq.it (M.P.); 2Microbiology Unit, Trieste University Hospital (ASUGI), 34125 Trieste, Italy; venera.costantino@asugi.sanita.fvg.it (V.C.); marina.busetti@asugi.sanita.fvg.it (M.B.); 3Infectious Diseases Unit, Trieste University Hospital (ASUGI), 34125 Trieste, Italy; omarsimonetti89@gmail.com (O.S.); roberto.luzzati@asugi.sanita.fvg.it (R.L.); 4Clinical Pathology and Microbiology Unit, “S. Giovanni di Dio” Hospital, 88900 Crotone, Italy; luigi.principe@gmail.com; 5Istituto Zooprofilattico Sperimentale dell’Abruzzo e del Molise, Campo Boario, 64100 Teramo, Italy; m.didomenico@izs.it (M.D.D.); a.rinaldi@izs.it (A.R.); c.camma@izs.it (C.C.)

**Keywords:** *Salmonella enterica* Goldcoast, *C. freundii*, WGS, VIM, IncA, β-lactamase, ARGs, virulence factors

## Abstract

In the present study, we analyzed the genome of two *S. enterica* strains TS1 and TS2 from stool and blood cultures, respectively, and one strain of *C. freundii* TS3, isolated from a single hospitalized patient with acute myeloid leukemia. The *S. enterica* Goldcoast ST358 (O:8 (C2-C3) serogroup), sequenced by the MiSeq Illumina system, showed the presence of β-lactamase genes (*bla*_VIM-1_, *bla*_SHV-12_ and *bla*_OXA-10_), *aadA1*, *ant(2″)-Ia*, *aac(6′)-Iaa*, *aac(6′)-Ib3*, *aac(6′)-Ib-cr*, *qnrVC6*, *parC(T57S)*, and several incompatibility plasmids. A wide variety of insertion sequences (ISs) and transposon elements were identified. In *C. freundii* TS3, these were the *bla*_VIM-1_, *bla*_CMY-150_, and *bla*_SHV-12_, *aadA1*, *aac(6′)-Ib3*, *aac(6′)-Ib-cr*, *mph(A)*, *sul1*, *dfrA14*, *ARR-2*, *qnrVC6*, and *qnrB38*. IncA plasmid isolated from *E.coli*/K12 transconjugant and *C. freundii* exhibited a sequence identity >99.9%. The transfer of IncA plasmid was evaluated by conjugation experiments.

## 1. Introduction

*Salmonella enterica* is an intracellular facultative anaerobe Gram-negative bacterium, belonging to the family of *Enterobacterales*. This common zoonotic pathogen is responsible for infections in both humans and animals [1]. *Salmonella* infection poses a major public health concern worldwide as a foodborne illness. Salmonellosis is the second most commonly reported gastrointestinal infection in the EU/EEA, and an important cause of food-borne outbreaks [2,3]. There are more than 2600 serovars of *Salmonella* on the basis of the lipopolysaccharide (LPS) O antigen and the flagellar protein (H) antigen [4]. From the clinical point of view, *S. enterica* subspecies *enterica* is the primary cause of human infections. *S. enterica* species can be subdivided into two big groups including the invasive typhoidal group (i.e., *S. Typhy*, *S. Paratyphy* A, B and C) and non-typhoidal *Salmonella*. Several hundred serovar of the *S. enterica* subspecies *enterica* are identified in different animal species (i.e., cattle, pig, poultry, human, etc.) [5,6]. Invasive typhoidal *Salmonella* causes enteric fever with clinical and severe complications if not treated [7]. Nontyphoidal *Salmonella* (NTS) strains include the *S. enterica* serovar Goldcoast, a rare zoonotic pathogen for human infection. The infection is usually acquired through contaminated food of animal origin [8]. *S. enterica* Goldcoast is a rare serovar for humans in the United States but it is more common in European countries. To date, a few epidemic outbreaks involving *S. enterica* Goldcoast have been described in Germany, Hungary, England, and Italy [9,10,11,12]. Recently, the increasing resistance to antimicrobials in Salmonella has posed a significant threat to public health. *Salmonella* strains resistant to ampicillin, trimethoprim-sulfamethoxazole, chloramphenicol, extended-spectrum cephalosporins, fluoroquinolones, tetracyclines and even carbapenems have emerged worldwide [1,13]. Β-Lactams are the most common antibiotics employed in clinical practice and their extensive use has led to the emergence of resistance. The World Health Organization (WHO) has estimated a high risk of the dissemination of antibiotic resistant bacteria (ARB) and an urgent worldwide action plan is required [14]. Carbapenems represent the “last-line agents” of defense for serious clinical infections caused by multidrug-resistant (MDR) Gram-negative bacteria and β-lactamases resistant to carbapenems (i.e., “carbapenemases”) are disseminated, mainly, among *Enterobacterales*. Currently, carbapenem-resistant *Enterobacterales* (CPEs) represent a serious problem in healthcare settings [14]. Resistance to carbapenems is due to the cooperation of different mechanisms including (i) production of carbapenemases, extended-spectrum-β-lactamases and AmpC, (ii) mutations that alter the expression and function of PBPs and porins, and (iii) efflux pump alteration [15]. The most worrisome carbapenemases found in *Enterobacterales* are represented by subclass B1 metallo-β-lactamases (MBLs) (i.e., NDM, VIM, and IMP), class A (i.e., KPC), and class D (i.e., OXA-48) enzymes of the Ambler classification. MBLs utilize one or two zinc ions for their catalysis while class A and D carbapenemases are serine-β-lactamases able to hydrolyze the β-lactams substrates by forming a transient or stable acyl enzyme through an active site serine. The dissemination of carbapenemase genes in bacterial isolates is due to their localization in mobile genetic elements which play a central role in facilitating horizontal genetic exchange and therefore promote the acquisition and spread of resistance genes. The most widespread CPEs worldwide in clinical settings are *Klebsiella pneumoniae* and *Escherichia coli* [16]. However, other CPEs causing nosocomial infections such as *Citrobacter* spp. and *Salmonella* spp. have, recently, been emerging [17,18]. Molecular epidemiology data on carbapenem-resistant *Citrobacter* spp. indicate the presence of class A and D carbapenemases (i.e., KPC-types and OXA-48, respectively) and MBLs (i.e., NDM-, VIM-, and IMP-variants) [17]. Even though carbapenems are not the drugs of choice for *Salmonella* infections, the presence of carbapenemases has been detected in NTS *S. enterica* identified in humans, animals, and food [19,20,21,22]. In particular, the first member of MBLs, the IMP-4 enzyme, was identified in a *S. waycross* isolate recovered in 2007 in Australia from an 87-year-old woman with a urinary tract infection followed by diarrhea [23]. However, the most common MBL reported worldwide in *S. enterica* is VIM-2, isolated for the first time in 2010 from the blood and urine of hospitalized patient in Morocco [24]. To date, the *bla*_VIM-1_ gene has mainly been identified in *S. enterica* strains isolated from livestock farms and food [25,26,27].

In the present study, we have reported a detailed characterization of a genome of two *S. enterica* (TS1 and TS2) strains and *C. freundii* TS3 isolated, from the stool and blood cultures of a single patient admitted to the University Hospital of Trieste (Northeastern Italy) with acute myeloid leukemia who underwent allogeneic stem cell transplantation complicated by graft-versus-host disease (GVHD). The IncA plasmid identified in both *S. enterica* and *C. freundii* was sequenced by Illumina and Nanopore systems in order to verify sequence similarity. Conjugation experiments were performed to support the theory of a possible “jump” in IncA plasmid or genetic determinants between *S. enterica* and *C. freundii*. Antibiotic resistance genes (ARGs), mobile genetic elements, and virulence factors were identified in both strains.

## 2. Results

### 2.1. Clinical Aspects and Strains Selection

We describe a patient with acute myeloid leukemia who underwent allogeneic stem cell transplantation (March 2022) complicated by graft-versus-host disease (GVHD). In late November 2022, the patient was hospitalized for acute diarrhea. Stool and blood cultures were positive for the third-generation cephalosporin-susceptible group C *S. enterica*. The strain was susceptible to carbapenems, cephalosporins, and aminoglycosides. In addition, *Clostridioides difficile* infection (CDI) was diagnosed. The patient was treated with fidaxomicin (10 days) and ceftriaxone (14 days) with clinical improvement and was discharged. Rectal colonization by KPC-producing *K. pneumoniae* and VIM-producing *C. freundii* was also detected. In late December 2022, the patient was readmitted for CDI recurrence that was treated with fidaxomicin (a tapered regimen for 20 days) and bezlotoxumab. The *S. enterica* group C was again identified from blood and stool cultures. Indeed, a second targeted antibiotic therapy with high-dose meropenem plus fosfomycin (14 days both) was administered. The Xpert Carba-R assay was carried out on *S. enterica* TS1, *S. enterica* TS2, and *C. freundii* TS3, and the presence of the *bla*_VIM_ gene was detected in these strains. The patient experienced a gradual improvement with symptoms resolution but remained hospitalized for GVHD management. A PET scan ruled out a deep-seated infection and bloodstream clearance was documented. In late January 2023, hospital-acquired pneumonia (HAP) was diagnosed. High-dose meropenem and colistin were started. Due to clinical deterioration and the recurrence of bloodstream infection (BSI) due to VIM-producing group C *S. enterica*, the antibiotic regimen was modified in high-dose tigecycline, cefiderocol, and colistin. The samples underwent genomic sequencing. Recurrent BSI due to infected gallstones and secondary HAP were hypothesized. After targeted therapy and removal of all intravascular devices, the blood cultures collected were negative. Unfortunately, the patient died of causes unrelated to the infection ten days later.

### 2.2. Antimicrobial Susceptibility of S. enterica and C. freundii

The *S. enterica* TS1 and *S. enterica* TS2 isolated from stool and blood cultures, respectively, were analyzed for their susceptibility profile against different antibiotics. Both strains showed resistance to cephalosporins and ciprofloxacin, susceptibility at increased exposure to imipenem, and susceptibility to meropenem, aminoglycosides, and tigecycline (Table 1). The *C. freundii* strain showed resistance to cephalosporins, susceptibility at increased exposure to imipenem, susceptibility to meropenem, aminoglycosides, ciprofloxacin, and tigecycline.

### 2.3. Genomic Features of S. enterica TS1 and TS2

Molecular analysis, performed by next-generation sequencing, of *S. enterica* TS1 and *S. enterica* TS2 showed the same genetic features (MLST, serovar, plasmids, ARGs, virulence factors, etc); therefore, from this point on we refer only to *S. enterica* TS2 isolated from the blood culture. The whole-genome sequence (WGS) of *S. enterica* TS2 consisted of 111 contigs, which comprise 5,094,694 bases. The overall G+C content amounted to 51.97%. According to the MLST Achtman scheme, which considers seven housekeeping genes (*aroC*, *dnaN*, *hemD*, *hisD*, *purE*, *sucA*, and *thrA*) the strain belonged to ST358. The *S. enterica* TS2 serovar, predicted by SeqSero 1.2, showed the Goldcoast serotype (antigenic formula: 8: r: l, w). *S. enterica* Goldcoast is a serovar of the O:8 (C2-C3) serogroup. The resistome of *S. enterica* Goldcoast TS2 showed a copious number of ARGs (Table 2). Concerning β-lactamases, the presence of the *bla*_VIM-1_, *bla*_SHV-12_, and *bla*_OXA-10_ determinants was ascertained. Resistance to aminoglycosides was represented by the *aadA1*, *ant(2″)-Ia*, *aac(6’)-Iaa*, *aac(6’)-Ib3*, and *aac(6’)-Ib-cr* determinants. The presence of *aac(6’)-Ib-cr* indicated simultaneous resistance to aminoglycosides and quinolones. However, resistance to quinolones was also mediated by the *qnrVC6* gene and it was chromosomally related to the mutation in *parC(T57S)*. Other ARGs responsible for resistance to macrolides (*mph(A)*), rifampicin (*ARR-2*), sulphonamides (*sul1*), tetracyclines (*tet(B)*), chloramphenicol (*catB8*), trimethoprim (*dfrA14*), and antiseptics (*qacE*) were also detected. The WGS data of *S. enterica* TS2 indicated that the isolate had several incompatibility plasmids such as IncA, IncHI2, IncHI2A, and IncFII(29). The pMLST showed that IncA and IncHI2 were represented by the ST12 and ST1 lineages, respectively. A wide variety of insertion sequences (ISs) and transposon elements were identified in the analyzed strain (Table 2). In detail, transposons Tn*6196* and Tn*1000* (Tn gamma delta) were detected. Sequence insertion elements such as IS*6100* of a length of 880 bp, IS*Ec52* of a length of 1250 bp, IS*903* of a length of 1057 bp, IS*As22b* of a length of 1198bp, IS*Sty2* of a length of 1259 bp, IS*421* of a length of 1340 bp, and MITEEc1 belonging to the IS*630* family were identified in the genome of *S. enterica* TS2. 

Six Salmonella pathogenicity islands (SPIs) (SPI-1, SPI-2, SPI-3, SPI-4, SPI-5, and SPI-13) and centisome 63 (C63PI) were found in the *S. enterica* Goldcoast genome. Several virulence genes were correlated with SPIs elements, such as *phoQ*, *sipA*, *sipB*, *sipC*, and *mgtB*. In SPI-1, the clusters *inv* (*invA*, *C*, *E*, and *G*) and *hilA*, *C*, *D* were also identified (Table 3). In SPI-2, *sseB*, *C*, *E*, and *D* genes were detected. Among conserved *Salmonella* fimbrial gene clusters (FGCs), *std*, *stb*, *stc*, *sth*, *bcf*, and *fim* were ascertained. The strain harbored the *cdtB* and *pltA* genes, related to the typhoid toxin, typical of *S. Typhi* and *S. Paratyphi*. *S. enterica* ST358 analyzed in this study also harbored the *pipB2*, *sseK2*, and *ssaI* virulence genes involved in intracellular proliferation, survival, and biofilm formation.

### 2.4. Genomic Features of C. freundii TS3

The *C. freundii* TS3 strain showed a draft genome of 5.251.986 bp and 91 contigs, with a G+C content (%) of 51.91%. Multi-locus sequence typing of *C. freundii* showed the ST62 lineage (http://pubmlst.org/organisms/citrobacter-spp/, accessed on 25 January 2023). Resistome analysis revealed the presence of genes encoding for resistance to aminoglycosides (*aadA1*, *aac(6′)-Ib3*, *aac(6′)-Ib-cr*), macrolides (*mph(A)*), sulphonamides (*sul1*), trimethoprim (*dfrA14*), antiseptics (*qacE*), and rifampicin (*ARR-2*). Quinolone resistance was mediated by *qnr* elements (*qnrVC6* and *qnrB38*), and the bi-functional *aac(6′)-Ib-cr* gene. The presence of *bla*_VIM-1_, *bla*_CMY-150_, and *bla*_SHV-12_ β-lactamases was also ascertained (Table 4). The *C. freundii* TS3 strain harbored an IncA plasmid pMLST ST12. As shown in Table 4, several ISs, such as IS*6100* (880bp), IS*5* (1195 bp), IS*Sen4* (1221 bp), and ISAs22 (1198 bp) were identified.

### 2.5. IncA Plasmid Mapping

In order to verify the sequence similarity of the IncA plasmid identified in *S. enterica* and *C. freundii*, the IncA plasmid was extracted from *C. freundii* TS3 and from transconjugant *E. coli* K12/*S. enterica* (see conjugation experiments in the materials and methods section). A total of 1.699.858.150 bp paired-end raw reads (510 Mb) were obtained by Illumina, and 126.920 raw reads with a mean length of 909.8 bp (115 Mb) by ONT for the *C. freundii* plasmid. The hybrid assembly produced 278 contigs with a largest contig of 150,373 bp. All contigs were used as inputs for MOB-suite to define their molecular type, only three were marked as plasmid sequences including the largest contig that results as a complete and circular plasmid. The IncA plasmids from transconjugant *E. coli K12/S. enterica* and *C. freundii* showed a sequence identity, by BLAST analysis, higher than 99.9%. The main genetic elements identified in the IncA plasmid were represented by plasmid replication module (*repA*), ARGs, IS family transposases, and genetic elements related to a conjugal transfer function. IncA harbored *bla*_VIM-1_ and *bla*_SHV-12_ which confer resistance to carbapenems and extended-spectrum β-lactams; *aacA4* is an aminoglycoside N-acetyltransferase that confers resistance to broad-spectrum aminoglycosides; *qnrVC6* confers resistance to fluoroquinolones; *ant1* is a streptomycin adenyltransferase; and the *mph(A)* gene is a macrolide phosphotransferase, which, mainly, inactivates erythromycin. A similarity search in a public sequence database indicated the presence, in the IncA plasmid, of 12 conjugation genes (*traB* to *traW*). The *traF*, *traG*, and *traH* genetic elements, identified as components of outer membrane complex, seem to be involved in conjugation transfer (Figure 1).

The IncA plasmid harbored a class 1 integron which includes (i) an integrase 1 gene from 117,171 bp to 116,158 bp, (ii) a variable region of about 4000 bp including five gene cassettes (*bla*_VIM-1_, *aacA4*, *qnrVC6*, *arr2*, and *ant1*), and (iii) a 3′CS region including *qacEdelta1* and *sul1* genes (Figure 2). Upstream and downstream of integron element, there are typical elements of transposon. 

### 2.6. IncA Transfer by Conjugation Assays

The transfer of IncA plasmid was evaluated by conjugation experiments. The selection of transconjugants was performed in plates supplemented with ceftazidime having ascertained the presence, in the IncA plasmid, of *bla*_VIM-1_ and *bla*_SHV-12_. Resistance to ceftazidime in *E. coli* K12 transconjugants was surely conferred by *bla*_SHV-12_. The correct transfer of the IncA plasmid was confirmed by amplification of the *bla*_VIM-1_ and *bla*_SHV-12_ genes in the *E. coli* K12 transconjugant of *S. enterica* and the *E. coli* K12 transconjugant of *C. freundii*. 

## 3. Discussion

Herein we describe the first identification in Italy of the VIM-1-producing *S. enterica* serovar Goldcoast ST358 isolated from the blood and stool cultures of a single patient colonized by multiple germs including *K. pneumoniae*, *C. difficile*, and *C. freundii* during a hospitalization for acute myeloid leukemia. The genome DNA of *S. enterica* (TS1 and TS2 strains) and *C. freundii* was sequenced and various ARGs were identified in the chromosome and plasmids of these clinical isolates. IncA was identified in both the *S. enterica* TS2 and *C. freundii* TS3 strains. In addition to IncA, *S. enterica* also possessed IncHI and IncFII plasmids. The complete sequence of incompatibility of the IncA plasmid, carrying *bla*_VIM-1_, recovered by both *S. enterica* TS2 and *C. freundii* TS3, showed a nucleotide sequence similarity higher than 99%, suggesting a possible transfer of IncA between the two isolates. The theory that *C. freundii* TS3 exchanged IncA plasmid with *S. enterica* TS2 or vice versa, during the patient’s hospitalization, was supported by the positive results of “in vitro” conjugation experiments and the presence of a genetic complex system subordinated to the conjugal function [28]. The IncA/C plasmids have an extremely broad host range coupled with a high ability to spread via conjugative transfer [29,30,31]. Conjugation is the most effective mechanism for horizontal gene transfer (HGT). In this context, various incompatibility plasmids have a huge impact on the global emergence of MDR bacteria in clinical, animal, and environmental settings [32,33]. Recently, a highly conjugative IncA plasmid has been characterized in several VIM-1-producing *Enterobacterales* [34]. Arcari and coworkers identified, in VIM-IncA, several genetic regions (IS*6000*, *dfrA14*, *mph(A)*, *bla*_SHV-12_, *aac(6′)-Ib3*, and *aadA1*) which were also encountered in our strains [34]. Despite the presence of *bla*_VIM-1_ in the IncA plasmid, both *S. enterica* TS2 and *C. freundii* TS3 were found to be susceptible to imipenem and meropenem. This is not unusual because several studies have reported low-carbapenem minimum inhibitory concentration values (MICs) in VIM-1-producing *Enterobacterales* [35,36]. In the chromosome of *S. enterica*, we have identified OXA-10, a class D oxacillinase commonly produced by *Pseudomonas aeruginosa*, which has been described as possessing weak activity against carbapenems [37]. The presence, in the IncA plasmid, of a class 1 integron with five gene cassettes has led to its rapid dissemination among bacteria, in particular when they are exposed to antibiotics. The IncA plasmid harbored the *bla*_SHV-12_ gene, whose presence also confers resistance to extended-spectrum β-lactams, and the fluoroquinolones and aminoglycoside resistance genes (*qnrVC6* and *aacA4*, respectively). *qnrVC6* is an emerging quinolone-resistance determinant in the *qnr* family which has mainly been identified in non-*Enterobacterales* species such as *Acinetobacter baumannii*, *Pseudomonas* spp., and *C. freundii* or, occasionally, in *Vibrio parahaemolyticus* [38,39,40,41]. In *S. enterica* TS2, resistance to fluoroquinolones is not only due to *qnrVC6* but also to *aac(6′)Ib-cr* and mutation in T57S amino acid substitution in *parC*. Mutation in *parC*, identified in the QRDR of *S. enterica* reduces susceptibility to ciprofloxacin even if ciprofloxacin is a drug of choice for treating *S. enterica* infections in humans [42]. In *S. enterica* TS2, the situation is exacerbated by the presence of virulence factors, in particular those belonging to fimbrial adherence determinants which are involved in the attachment of *Salmonella* cells to the epithelium of gut. The majority of virulence genes are positioned within SPIs on chromosome or virulence plasmids. To date, 21 SPIs have been identified in *Salmonella* spp. [43]. 

The interesting point of the present study is the finding of *bla*_VIM-1_-producing *S. enterica* Goldcoast in a bloodstream infection. Indeed, the *S. enterica* serovar Goldcoast is an NTS-causing zoonotic foodborne disease and it was isolated for the first time in Ghana in 1953 [44]. Based on a European Food Safety Authority (EFSA) report in the 2020–2021 period, carbapenemase-producing *Salmonella* have not been detected in humans and food-producing animals [45]. After the first isolation of *S. enterica* Goldcoast, sporadic cases have been identified in Europe. Nevertheless, since 1984–1985, *S. enterica* Goldcoast started to be isolated in Europe more frequently. An outbreak of *S. enterica* Goldcoast, recorded in 2009–2010 in Italy, as well as in other European countries, was caused by widespread consumption of pork-containing products [12]. In 2013, *S. enterica* Goldcoast, associated with whelk consumption, caused gastroenteritis in a large number of people in England [11]. By 2019, *S. enterica* Goldcoast was also reported in China and Taiwan [46,47,48]. The main drugs of choice for *S. enterica* treatment are broad-spectrum cephalosporins and fluoroquinolones, while carbapenems are reserved for severe infections. Nevertheless, resistance to carbapenems represents a real public health problem. The main carbapenemases identified in *Salmonella* from humans and food-producing animals are KPC-2 (class A), OXA-48 (class D), and MBLs in particular NDM-1, NDM-5, VIM-2, and IMP-4 [18]. In 2014, Sotillo and coworkers reported the emergence of the VIM-1-producing *S. enterica* serovar Typhimurium in a pediatric patient [49]. Since then, the *bla*_VIM-1_ gene was found in the *S. enterica* serovar Infantis isolated in swine and poultry farms in Germany [26,50]. To date, the β-lactamases identified in *S. enterica* Goldcoast are TEM-1, CTX-M-55, OXA-48, AAC-1, and VIM-1 [25,46,51]. Based on an EFSA report (2021), 5.1% of *S. enterica* Goldcoast harbor the AmpC β-lactamases, in particular CMY-2 [2], which we did not detect in the strains analyzed in the present study. Analyzing the genome of *C. freundii* TS3, it has many ARGs in common with *S. enterica* Goldcoast TS2 (*aadA1*, *aac(6′)-Ib3*, *aac(6′)-Ib-cr*, *sul1*, *dfrA14,* and *qacE*). Unlike *S. enterica* TS2, *C. freundii* TS3 showed the presence of CMY-150, a CMY-2-variant, which confers resistance to ceftazidime and aztreonam [52]. Carbapenemase-producing *Citrobacter* spp. is particularly increasing in some geographical regions. The typical sequence types of *Citrobacter* spp., which have emerged as the dominant clones in healthcare settings, are ST19 and ST22 [53]. In Italy, the majority of VIM-1-producing *C. freundii* belong to ST22, ST112, ST523, and ST686 [17]. 

## 4. Materials and Methods

### 4.1. Strains Selection 

Periodic screening rectal swabs were analyzed for the presence of MDR Gram-negative pathogens by selective chromogenic plates (Agar chromID CARBA SMART, bioMérieux, Marcy l’Étoile, France). Blood cultures were performed by BactAlert (bioMérieux, Marcy l’Étoile, France); positive bottles were subcultured on blood agar and selective chromogenic plates (Agar chromID CPS Elite, bioMérieux, Marcy l’Étoile, France). In diarrhoeic stools, the presence of *Salmonella* was detected on selective chromogenic plates (Agar chromID Salmonella Elite, BioMerieux). *S. enterica* typing was performed by serological tests (SeroQuick Id kit, SSI Diagnostica, Hillerød, Denmark). The presence of toxin-producing *Clostridioides difficile* was detected by molecular testing (Xpert *C. difficile* BT assay, Cepheid, Sunnyvale, CA, USA). The colonies were identified by matrix-assisted laser desorption ionization time-of-flight mass spectrometry (MALDI-TOF) VitekMS (bioMérieux, Marcy l’Étoile, France). To detect carbapenem-resistant genes (*bla*_VIM_, *bla*_IMP_, *bla*_NDM_, *bla*_KPC_, and *bla*_OXA-48_), a molecular biology test was carried out by Xpert Carba-R assay (Cepheid, Sunnyvale, CA, USA).

### 4.2. Antimicrobial Susceptibility Testing

Antimicrobial susceptibility testing for Gram-negative organisms was carried out by broth microdilution test (Sensititre, ThermoFisher, Waltham, MA, USA) according to the manufacturer’s instructions. The following antimicrobial agents were tested: amikacin, cefotaxime, ceftazidime, cefepime, ciprofloxacin, gentamicin, imipenem, meropenem, and tigecycline. MICs were interpreted according to EUCAST 2022 criteria (https://www.eucast.org/clinical_breakpoints, accessed on 10 November 2022). 

### 4.3. Genomic and Plasmid Extraction

Total nucleic acid was extracted from liquid cultures of *S. enterica* (TS1 and TS2 strains) and *C. freundii* TS3 using a modified protocol of the MagMAX Microbiome Ultra Nucleic Acid Isolation kit (Applied Biosystems, ThermoFisher Scientific, Monza, Italy) as previously reported [54]. The IncA plasmid was extracted from *S. enterica* TS2 and *C. freundii* TS3 using a Large-Construct Kit, which allows a purified plasmid free of genomic DNA contamination to be obtained.

### 4.4. Whole-Genome Sequencing (WGS) and Bioinformatic Analysis

Short-read sequencing libraries were prepared with an Illumina DNA Prep Kit (Illumina Inc. San Diego, CA, USA) and sequenced on an Illumina MiSeq instrument with a 2 × 300 bp paired-end protocol as previously described [55,56]. Quality control and sequence filtering were performed using DRAGEN FastQC + MultiQC v3.6.3 (https://basespace.illumina.com/apps/10562553/DRAGEN-FastQC-MultiQC, accessed on 21 January 2023). Paired-end reads were assembled with Velvet v1.2.10 (https://basespace.illumina.com/apps/8556549/Velvet-de-novo-Assembly, accessed on 21 January 2023). 

### 4.5. WGS-Based AMR Identification and Serotype Prediction of S. enterica TS2

ResFinder 4.1 and PlasmidFinder 2.0 were used to detect acquired ARGs and plasmids, respectively (https://www.genomicepidemiology.org/services/, accessed on 25 January 2023). The genome was also assigned to ST using MLST 2.0 (https://cge.cbs.dtu.dk/services/MLST/, accessed on 25 January 2023). Identified plasmids of the IncF, IncH1, IncH2, IncI1, IncN, or IncA/C type were subtyped by pMLST 2.0 (https://cge.food.dtu.dk/services/pMLST/, accessed on 25 January 2023). Serotyping of *Salmonella* spp. was predicted by using WGS-assembled contig files (in FASTA format) with SeqSero1.2 (https://cge.cbs.dtu.dk/services/SeqSero/, accessed on 25 January 2023). Virulence factors and Salmonella Pathogenicity Islands were detected using the Virulence Factor Database (VFDB) (http://www.mgc.ac.cn/VFs/, accessed on 26 January 2023) and SPIFinder 2.0 (https://cge.food.dtu.dk/services/SPIFinder/, accessed on 25 January 2023), respectively. MobileElementFinder v.1.0.3 was used to identify mobile genetic elements (https://cge.food.dtu.dk/services/MobileElementFinder/, accessed on 25 January 2023).

### 4.6. WGS-Based AMR Identification of C. freundii TS3

ARGs, plasmids, mobile genetic elements, MLST and pMLST were detected as described in the above section, using services provided by Center for Genomic Epidemiology (https://www.genomicepidemiology.org/, accessed on 25 January 2023).

### 4.7. IncA Plasmid Sequencing

Plasmid DNA isolated from *C. freundii* was sequenced by both Illumina and Oxford Nanopore technologies (ONT) on NextSeq2000 setting 150 PE reads, and GridION using FLO-MIN112, respectively. Library preparation was accomplished using the DNA library prep (Illumina) and Native Barcoding Kit 24 SQK-NBD112.24 (ONT). Hybrid assembly was obtained using Unicycler v0.5.0 [57] combining short and long reads. Plasmid sequence was analyzed through the MOB-suite v.3.0.3 [58] which contains software tools for clustering, reconstruction, and typing of plasmids from draft assemblies. The plasmid was annotated by PROKKA v.1.14.5 [59] using the reference sequence FJ705807 [60]. Plasmid DNA isolated from *S. enterica* was sequenced by Oxford Nanopore technologies on GridION using the Native Barcoding Kit 24 SQK-NBD112.24 (ONT), and raw reads were assembled by Flye v.2.9.2 [61]. Finally, the sequence similarity of IncA plasmids was evaluated by blasting (https://blast.ncbi.nlm.nih.gov/Blast.cgi?PAGE=MegaBlast&PROGRAM=blastn&PAGE_TYPE=BlastSearch&BLAST_SPEC=blast2seq, accessed on 25 March 2023) the draft plasmid sequence of *S. enterica* vs. the complete *C. freundii* plasmid. 

### 4.8. Conjugation Experiments

*S. enterica* and *C. freundii* and *E. coli* K12 strains were used as donors and recipient for conjugation experiments, respectively. Transconjugants were selected on Luria-Bertani (LB) agar plates supplemented with 250 mg/L of streptomycin and 32 mg/L ceftazidime. The detection sensitivity of the assay was ≥5 × 10^−7^ transconjugants per recipient.

## 5. Conclusions

The increase in resistance to β-lactams is a result of the high plasticity of β-lactamases genes and the genetic adaptability of the bacterial population in a short period of time. The presence, in the IncA plasmid, of genetic components involved in conjugation (i.e., *traF* and *traH*) facilitates the genetic transfer of ARGs [28]. Carbapenem-resistant *S. enterica* Goldcoast and *C. freundii* are not considered to be a classical nosocomial pathogen but they possessed many genetic elements which conferred high resistance to several antibiotic families. Bloodstream *S. enterica* infections are very common and represent a real risk for public health, in particular when multiple bacterial species colonize the same patient [62]. The therapeutic options are very limited and the patient risks their own life.

## Figures and Tables

**Figure 1 antibiotics-12-01659-f001:**
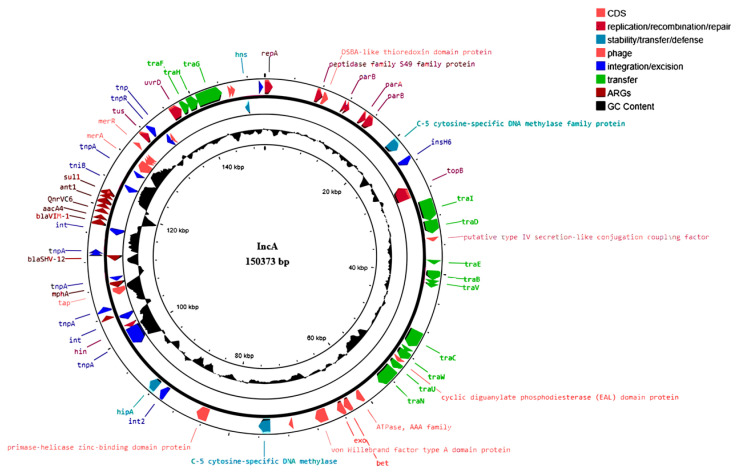
Circular map of IncA plasmid. The circles, from the outermost to the innermost, show: (i) coding DNA sequences (CDSs) encoded on the plus and minus DNA strands (annotation is reported for known genes only); and (ii) the G+C content, shown as deviation from the average G+C content of the entire molecule. Genes encoding protein of known functions are in different colors, as detailed in the legend.

**Figure 2 antibiotics-12-01659-f002:**
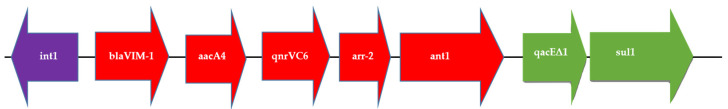
Structure of the class 1 integron carried on the analyzed plasmid. The *intI1* integrase gene is indicated by a violet arrow; the variable region includes five gene cassettes (red arrows).

**Table 1 antibiotics-12-01659-t001:** Antimicrobial susceptibility pattern of *S. enterica* and *C. freundii* clinical strains.

Antibiotics	MIC (mg/L)			
	*S. enterica* TS1, TS2		*C. freundii* TS3	
Cefotaxime	>4	(R)	>4	(R)
Ceftazidime	>128	(R)	>128	(R)
Cefepime	>16	(R)	>16	(R)
Imipenem	4	(I)	4	(I)
Meropenem	2	(S)	1	(S)
Ciprofloxacin	0.5	(R)	0.12	(S)
Amikacin	≤4	(S)	≤4	(S)
Gentamicin	2	(S)	2	(S)
Tigecycline	0.25	(S)	0.25	(S)

**Table 2 antibiotics-12-01659-t002:** ARGs and mobile genetic elements in the *S. enterica* Goldcoast TS2 strain.

Genotyping of *S. enterica* Goldcoast TS2 Strain						
MLST	Β-Lactam Resistance	Aminoglycoside Resistance	Macrolide Resistance	Quinolone Resistance	Rifampicin Resistance	Others
ST358	*bla* _VIM-1_ *bla* _OXA-10_ *bla* _SHV-12_	*aadA1* *ant(2″)-Ia* *aac(6′)-Iaa* *aac(6′)-Ib3* *aac(6′)-Ib-cr*	*mph(A)*	*qnrVC6* *parC T57S* *aac(6′)-Ib-cr*	*ARR-2*	*sul1* *tet(B)* *catB8* *dfrA14* *qacE*
**Mobile genetic elements**						
**Transposons**		**Plasmids**	**pMLST**		**Insertion Sequences (ISs)**	
Tn*1000* Tn*6196*		IncA, IncHI2, IncHI2A, IncFII(29)	IncA:(ST = 12) IncHI2:(ST = 1)		IS*6100* (IS*6* family) IS*Ec52* IS*903* IS*As22b* (IS*3* family; IS*407* group) MITEEc1 (IS*630* family) IS*Sty2* (IS*3* family) IS*421* (IS*4* family; IS*231* group)	

**Table 3 antibiotics-12-01659-t003:** Virulence factors (VFs) in the *S. enterica* Goldcoast TS2 strain.

VF Class	Virulence Factors	Related Genes
Fimbrial Adherence Determinants	Agf/Csg	*csgG*
	Bcf	*bcfB*,*C*,*D*
	Fim	*fimC*,*D*,*H*,*Z*
	Saf	*safB*,*C*
	Stb	*stbB*,*C*,*D*
	Stc	*stcC*,*D*
	Std	*stdB*,*C*
	Ste	*steB*,*C*
	Stf	*stfC*,*D*
	Sth	*sthC*,*E*
	Sti	*stiC*,*H*
Macrophage Inducible Genes	Mig-14	*mig-14*
Magnesium Uptake	Mg2+ transport	*mgtB*
Nonfimbrial Adherence Determinants	MisL	*misL*
	RatB	*ratB*
	SinH	*sinH*
Regulation	PhoPQ	*phoQ*
Secretion System	TTSS (SPI-1 encode)	*hilA*,*C*,*D*; *invA*,*C*,*E*,*G*; *prgH*; *sipD*; *spaO*,*R*,*S*;
	TTSS (SPI-2 encode)	*ssaC*,*D*,*L*,*N*,*Q*,*U*,*V*; *sseC*; *ssrA*
	TTSS effectors translocated via both systems	*slrP*
	TTSS-1 translocated effectors	*avrA*; *sipA*,*B*,*C*; *sopA*,*B*,*D*; *sptB*
	TTSS-2 translocated effectors	*pipB2*; *pipB*; *sifA*,*B*; *sopD2*; *sseF*,*G*,*J*,*K1*,*K2*,*L*
Toxin	Typhoid toxin	*cdtB*, *pltA*

**Table 4 antibiotics-12-01659-t004:** ARGs and mobile genetic elements of the *C. freundii* TS3 strain.

Genotyping of *C. freundii* TS3						
MLST	Β-Lactam Resistance	Aminoglycoside Resistance	Macrolide Resistance	Quinolone Resistance	Rifampicin Resistance	Others
ST62	*bla* _VIM-1_ *bla* _CMY-150_ *bla* _SHV-12_	*aadA1* *aac(6′)-Ib3* *aac(6′)-Ib-cr*	*mph(A)*	*qnrVC6* *qnrB38* *aac(6′)-Ib-cr*	*ARR-2*	*sul1* *dfrA14* *qacE*
**Mobile genetic elements**						
**Plasmids**		**pMLST**	**Insertion Sequences (ISs)**			
IncA		IncA:(ST = 12)	IS*6100* (IS*6* family), IS*5* (IS*5* family), IS*Sen4* (IS*3* family), IS*As22* (IS*3* family)			

## Data Availability

Data are contained within the article.

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
