# Peer review of "Identification of IncA Plasmid, Harboring blaVIM-1 Gene, in S. enterica Goldcoast ST358 and C. freundii ST62 Isolated in a Hospitalized Patient"

_antibiotics, 2023, doi:10.3390/antibiotics12121659_

Round 1
Reviewer 1 Report
Comments and Suggestions for Authors
I have reviewed a paper titled "Identification of IncA Plasmid, harbouring balVIM-1 gene, in S.enterica Goldcoast ST358 and C. freundii ST62 isolated in a hospitilized patient". In my view, this study is relevant to the scope of the journal. However, the following points need to be addressed:
1. Given that the study involves humans, the Ethics certificate number must be included in the study.
2. Figures 1 and 2 appear to be suboptimal. Please improve the visualization of those figures.
Author Response
Response to Reviewer 1
The authors thank reviewer 1 for careful revision of the manuscript.
- The study was conducted in the context of normal clinical routine. All information about the patient has been anonymized. Samples were coded and analyzed with an anonymized database.
- Figures 1 and 2 have been performed and given in .png
Reviewer 2 Report
Comments and Suggestions for Authors
The manuscript presents the characterization of the IncA plasmid, harboring an important beta-lactam antibiotic resistance gene, blaVIM-1 gene, in S. enterica Goldcoast and C. freundii isolated from one patient. An interesting clinical case, scientifically correct research, and a concisely written manuscript make it a pleasure to read.
Minor comments concern only the editorial side:
Line 110 and 112: the abbreviation BSI has no explanation
Figures need to have better resolution.
Author Response
Response to Reviewer 2
The authors thank reviewer 2 for careful revision of the manuscript.
- Line 110 and 112. The BSI abbreviation has been explained.
- Figures 1 and 2 have been performed and given in png file at 300 dpi of resolution.
Reviewer 3 Report
Comments and Suggestions for Authors
The manuscript, titled "Identification of IncA Plasmid Harboring the blaVIM-1 Gene in S. enterica Goldcoast ST358 and C. freundii ST62 Isolated from a Hospitalized Patient," is not only well-designed but also excellently written.
Given the inclusion of "S. enterica Goldcoast" in the manuscript title, the introduction should provide an overview of prior research related to the sources and resistance determinants specific to this serovar or sequence types.
Comments on the Quality of English LanguageThe English quality is fine. Minor gramatical or syntax errors
Author Response
Response to Reviewer 3
The authors thank reviewer 3 for careful revision of the manuscript.
The introduction has been improved and a couple of references has been added.